# The Myosin-V Myo51 and Alpha-Actinin Ain1p Cooperate during Contractile Ring Assembly and Disassembly in Fission Yeast Cytokinesis

**DOI:** 10.3390/jof10090647

**Published:** 2024-09-12

**Authors:** Zoe L. Tyree, Kimberly Bellingham-Johnstun, Jessica Martinez-Baird, Caroline Laplante

**Affiliations:** 1Molecular Biomedical Sciences Department, College of Veterinary Medicine, North Carolina State University, Raleigh, NC 27607, USA; zltyree@ncsu.edu (Z.L.T.); ksbellin@ncsu.edu (K.B.-J.); jlmart13@ncsu.edu (J.M.-B.); 2Quantitative and Computational Developmental Biology Cluster, North Carolina State University, Raleigh, NC 27607, USA

**Keywords:** cytokinesis, Myosin-V, alpha-actinin

## Abstract

Cytokinesis is driven in part by the constriction of a ring of actin filaments, myosin motors and other proteins. In fission yeast, three myosins contribute to cytokinesis including a Myosin-V Myo51. As Myosin-Vs typically carry cargo along actin filaments, the role of Myo51 in cytokinesis remains unclear. The previous work suggests that Myo51 may crosslink actin filaments. We hypothesized that if Myo51 crosslinks actin filaments, cells carrying double deletions of *ain1*, which encodes the crosslinker alpha-actinin, and *myo51* (*∆ain1 ∆myo51* cells) will exhibit more severe cytokinesis phenotypes than cells with the single *∆ain1* mutation. Contrary to our expectations, we found that the loss of Myo51 in *∆ain1* cells partially rescued the severity of the node clumping phenotype measured in *∆ain1* cells. Furthermore, we describe a normal process of contractile ring “shedding”, the appearance of fragments of ring material extending away from the contractile ring along the ingressing septum that occurs in the second half of constriction. We measured that *∆ain1 ∆myo51* cells exhibit premature and exaggerated shedding. Our work suggests that Myo51 is not a simple actin filament crosslinker. Instead, a role in effective node motion better recapitulates its function during ring assembly and disassembly.

## 1. Introduction

The contractile ring of actin and myosin supports cytokinesis, the physical separation of the cytoplasm of a mother cell into two daughter cells. Myosin motor proteins produce contractile forces in cells by binding to and pulling on actin filaments. In fission yeast, three myosins cooperate during cytokinesis: the two Myosin-II, Myo2 and Myp2 (composed of either heavy chain Myo2p or Myp2p and their light chains Rlc1p and Cdc4p), and the Myosin-V, Myo51 (composed of the heavy chain Myo51p, light chains Cam1p and Cdc4p, and accessory proteins Rng8p and Rng9p) [1]. Cells with mutations in pairs of these three myosins show that Myo2 is most important for ring assembly, while Myp2 is most important during constriction [1]. Myo51 supports both type-II myosins in their respective roles. How a Myosin-V contributes to cytokinesis remains an unanswered question. Electron microscopy and hydrodynamic measurements of recombinant Myo51p expressed and purified from insect cells show that Myo51 bound by Rng8p/9p is a single headed Myosin-V that interacts with actin filaments both at its motor and tail domains in vitro, suggesting that it may crosslink actin filaments or transport actin filaments within the contractile ring [2,3]. Even with this important molecular information about Myo51, how Myo51 supports the role of Myo2 and Myp2 during contractile ring assembly and constriction remains unclear.

The contractile ring is a complex cellular machine with a largely unknown molecular organization. Understanding how proteins organize inside the functional contractile ring is key to uncovering the mechanism of cytokinesis. In fission yeast, the contractile ring assembles by the coalescence of nodes, protein complexes composed of Cdc12p, the FBAR protein Cdc15p, the IQGAP homolog Rng2p and the Myosin-II molecule Myo2, into a continuous contractile ring by a search capture pull and release (SCPR) mechanism. During this mechanism, an actin filament polymerized by Cdc12p in one node is captured by Myo2 in a neighboring node [4]. Myo2 then pulls on the actin filament, bringing the nodes together. Actin filament connections between nodes must be released by severing from cofilin/Adf1p to prevent the formation of clumps of nodes [4,5]. Simulations of this minimalistic SCPR mechanism recapitulate the coalescence of nodes into a contractile ring [4]. The addition of actin filament crosslinkers recapitulated the alignment of nodes into “linear structures”, seen in wild-type and *cdc25-22* arrested and released cells, and enhanced the robustness of the SCPR mechanism of node coalescence [6,7]. The experimental work supports the importance of actin filament crosslinkers alpha-actinin/Ain1p and fimbrin/Fim1p in the alignment of the nodes onto a network of bundles of actin filaments. Ain1p and Fim1p prevent the clumping of nodes and ensure the assembly of a continuous ring without gaps [6]. Interestingly, cells that lack Myo51p (*∆myo51* cells) exhibit delayed node coalescence and an increased population of immobile nodes, suggesting that Myo51 alone may help engage nodes with the underlying actin filament network [1,2].

Advanced microscopy techniques, including single molecule localization microscopy (SMLM) and electron cryotomography (ECT), are beginning to uncover the molecular organization of the contractile ring [8,9,10,11]. SMLM revealed the molecular organization of nodes with a core of the node containing Cdc15p, Cdc12p, Rng2p and the tips of the Myo2 tails positioned against the cytoplasmic face of the plasma membrane and the heads of Myo2 fanning out into the cytoplasm like an inverted bouquet [9]. This molecular organization, along with biochemical, cellular biology and mechanical probing, suggest that nodes likely anchor the contractile ring to the plasma membrane and cell wall [8,9,10,12,13,14,15,16]. ECT in fission yeast cells revealed the organization of the actin filaments in the constricting ring [11]. The main bundle of actin filaments is located ~100 nm on the cytoplasmic side of the plasma membrane. Actin filaments within the main bundle are mostly parallel to each other and to the plasma membrane. Actin filament binding proteins, including the passive crosslinker Ain1p, likely organize the actin filaments of the contractile ring into a bundle. About 250 dimers of Ain1p localize to the full-size contractile ring in fission yeast [17]. *∆ain1* cells exhibit a contractile ring assembly defect where nodes coalesce into clumps rather than a uniform contractile ring [5,6]. Other proteins likely crosslink actin filaments and compensate for the function of Ain1p as *∆ain1* cells show no defect in contractile ring constriction [6]. Ain1p and Fim1p compete for actin filament binding and in *∆ain1* cells, Fim1p, a crosslinker typically restricted to actin patches, joins the contractile ring [18,19]. Therefore, Fim1p may compensate for the lack of Ain1p in *∆ain1* cells. However, the presence of Fim1p in the contractile ring may also cause alteration of the organization of the actin filament network of the contractile ring owing to differences in the molecular structure of the two crosslinkers.

Here, we investigated the potential role of Myo51 as an actin filament crosslinker during cytokinesis. We hypothesized that if Myo51 crosslinks actin filaments, cells carrying double deletions of *ain1* and *myo51* (*∆ain1 ∆myo51* cells) will exhibit more severe cytokinetic phenotypes than *∆ain1* cells. Similarly to Ain1p, there are ~600 polypeptides of Myo51p in a full-size contractile ring [6,17]. Myo51 localized to the inner layer of the contractile ring in an actin-filament-dependent manner during cytokinesis. As previously demonstrated, the loss of Ain1p resulted in clumps of mEGFP-Myo2p forming during ring assembly [6]. Unexpectedly, deleting *myo51* in *∆ain1* cells partially rescued the severity of the clumping phenotype, suggesting that Myo51 does not simply crosslink actin filaments. We also noticed that constricting contractile rings in *∆ain1 ∆myo51* cells showed ring material extending away from the contractile ring along the ingressing septum. This normal process that we named “shedding” appears to be a mode of contractile ring disassembly that begins during the second half of constriction when the ring is at least 50% constricted. We measured that *∆ain1 ∆myo51* cells exhibit premature and exaggerated shedding, suggesting a role for Myo51 and Ain1p during shedding and perhaps ring disassembly. Our work suggests that Myo51 is not a simple actin filament crosslinker. Instead, Myo51 may act to effectively engage nodes with the actin filament network, resulting in efficient node motions during ring assembly and disassembly.

## 2. Materials and Methods

### 2.1. Strains, Growing Conditions and Genetic and Cellular Methods

Appendix A lists the *S. pombe* strains described in this study. The strains were created using PCR-based gene targeting to integrate the constructs into the locus of choice and confirmed by PCR, sequencing and fluorescence microscopy [20]. Primers with 80 bp of homologous sequence flanking the integration site (obtained at https://bahlerlab.info/resources/ (accessed on 1 December 2023)) and two repeats of GGAGGT to create a 4xGly linker were used to amplify the vector of choice. The cells were grown in the exponential phase at 25 °C for 36–48 h in YE5S-rich liquid medium in 50-mL baffled flasks in a shaking incubator in the dark before imaging. To synchronize the population of cells, we used the temperature-sensitive *cdc25-22* mutation to arrest cells at the G2/M transition at the restrictive temperature of 36 °C for 4 h. We then released cells into mitosis at the permissive temperature of 22 °C as a synchronized population. The cells were concentrated 10- to 20-fold by centrifugation at 2400× *g* for 30 s and then resuspended in EMM5S. Then, 5 μL of cells was mounted on a thin gelatin pad consisting of 10 μL 25% gelatin (Sigma-Aldrich, St. Louis, MO, USA; G-2500) in EMM5S, sealed under a #1.5 coverslip with VALAP (1:1:1 Vaseline:Lanolin:Parafin), and observed at 22 °C. The cells were grown in the exponential phase at 25 °C in YE5S-rich liquid medium in 50-mL baffled flasks in a shaking incubator in the dark.

To image cells vertically in microfabricated yeast holders (yeast motels), 5 μL of a diluted resuspension of yeast cells as described above was pipetted onto the surface of a polydimethylsiloxane mold containing wells 6 or 7 µm in diameter and 14 µm in depth [1]. The mold was then inverted onto a 40 mm in diameter circular #1 coverslip, and the cells were imaged immediately.

To partially depolymerize actin, the cells were treated with 2.5 µM LatA for 6 min before imaging, mounted as described below and imaged immediately.

The comparison of growth between strains was performed with a ten-fold dilution assay of liquid cultures grown in YE5S for 36 h and diluted to maintain OD_595_ between 0.05 and 0.5. The starting culture for the dilution series was at OD_595_ 0.2, and each following culture was diluted 10 times. Five microliters of each serial dilution was spotted on YE5S plates and incubated for 48 h at 25, 32 or 36 °C. Duplicate plates were prepared for each temperature tested.

### 2.2. Spinning-Disk Confocal Microscopy

For imaging, the cells were prepared as described above. Fluorescence images of live cells were acquired with a Nikon Eclipse Ti microscope equipped with a 100×/numerical aperture (NA) 1.49 HP Apo TIRF objective (Nikon, Minato City, Japan), 405/561 nm solid state lasers and an electron-multiplying cooled charge-coupled device camera (EMCCD IXon 897; Andor Technology, Belfast, UK) using Nikon Element software (AR 5.02.01, Nikon, Minato City, Japan). The Nikon Element software was used for acquisition.

### 2.3. Super-Resolution Data Acquisition and Display

Super-resolution imaging was performed as described previously [9]. The acquired data were processed to localize single molecules as previously described [8,9,21]. The acquired frames were analyzed using a custom sCMOS-specific localization algorithm based on a maximum likelihood estimator (MLE), as described previously [8,9,21,22]. A log-likelihood ratio was used as the rejection algorithm to filter out overlapping emitters, nonconverging fits, out-of-focus single molecules and artifacts caused by rapid movements during one camera exposure time [22,23]. The accepted estimates were reconstructed in a 2D histogram image of 5-nm pixels, where the integer value in each pixel represented the number of localization estimates within that pixel. The images for visualization purposes were generated with each localization convolved with a 2D Gaussian kernel (σ = 7.5 nm). The images were reconstructed from all or a subset of acquired frames and color-coded for either the temporal information (JET LUT map) or for localization density (Heat LUT map). Our localization algorithm eliminated out-of-focus emissions, providing an effective depth of field of ~400 nm [9].

### 2.4. Measurement of Contractile Ring Timing

Contractile ring timing was measured by noting the frame in the timelapse micrograph where the spindle pole body (SPB) separated, the contractile ring completed assembly, constriction onset occurred, the contractile ring began shedding, and the contractile ring completed disassembly. If the clumping phenotype was observed, the clumping start and end frames were also noted. The completion of contractile ring assembly was defined as the timepoint at which all material had fully incorporated into a single contractile ring structure. The constriction onset was defined as the timepoint at which the contractile ring began to consistently decrease in diameter on a kymograph. The beginning of contractile ring shedding was defined as the time point at which material could be seen dissociating from the ring in a consistent manner. The completion of contractile ring disassembly was defined as the timepoint at which the signal from the disassembling contractile ring was no longer visible. The beginning of ring clumping was defined as the point at which a dense persistent region of material was formed, creating a nonuniform ring. The end of ring clumping was defined as the time point at which material was observed uniformly across the ring. Log rank tests were used to determine whether there was a significant difference between genotypes (Appendix A).

### 2.5. Classification of Clumping and Shedding Phenotypes

A ring with mild clumping refers to a contractile ring with no gaps in signal and a mild contrast between stretches of brighter and dimmer fluorescent signal. A ring with severe clumping was defined by a discontinuous fluorescent signal with large gaps in the ring or stretches of intense mEGFP-Myo2p signal separated by dim signal.

Exaggerated shedding was defined as shedding fragments that are long enough to reach the width of the cell and bright enough to obscure the body of the contractile ring, making it difficult to distinguish the contractile ring from the shedding fragment.

### 2.6. Measurement of Clumping Fluorescence Intensity

Kymographs of rings expressing a clumping phenotype were created and thresholded in ImageJ/Fiji (2.14.0/1.54f, National Institutes of Health, Bethesda, MD, USA) to highlight the region of highest fluorescence caused by the clumping of mEGFP-Myo2p signal [24]. The mean gray value of this region was used as the fluorescence of the clumping region. To calculate the background fluorescence of the clumped region, a line was drawn from the top to the bottom edge of the kymograph using a line width equal to the size of the clumped region. The mean gray value of this region was used as the background fluorescence. The final fluorescence of the region was calculated by subtracting the background fluorescence from the fluorescence of the clumped region. Student’s *t* tests were used to determine whether there was a significant difference in the fluorescence intensity of the clumped regions.

### 2.7. Measurement of Global Cytoplasmic Fluorescence Intensity

To measure global cytoplasmic fluorescence intensity, sum projections of stacks of 21 optical images separated by 0.36 µm were created from confocal micrographs of cells treated with DMSO or 2.5 µM LatA. Cells of similar size were selected for this analysis. For each cell measured, a region of interest was created with the polygon tool in ImageJ/Fiji (2.14.0/1.54f, National Institutes of Health, Bethesda, MD, USA), encompassing the entire cell, and the total fluorescence intensity within that polygon was measured with the integrated density. Student’s *t* tests were used to determine whether there was a significant difference in the fluorescence intensity between the DMSO and LatA populations.

### 2.8. Measurement of Length of Shedding Fragments

The length of the shedding fragments extending from the contractile rings was measured in kymographs. The total width of the cell was estimated as the diameter of the contractile ring before the onset of ring constriction. Student’s *t* tests were used to determine whether there was a significant difference in the length of shedding fragments between the populations.

### 2.9. Reporting Summary Statistics

Appendix A lists all the means, standard deviations and sample sizes for all the data reported in swarm plots.

## 3. Results

### 3.1. Loss of Myo51 Partially Rescues the ∆ain1 Node Clumping Phenotype during Ring Assembly

Myo51 can bind actin filaments at both its motor domain and tail, suggesting that it may act as a crosslinker [3]. Myo51 may therefore cooperate with the passive actin filament crosslinker Ain1p during cytokinesis. It was previously shown by Laporte et al. that *∆ain1* cells expressing other node markers Rlc1p, Cdc15p, Myo2p and Mid1p exhibit clumping [6]. We acquired timelapse confocal micrographs of *∆ain1* cells expressing mEGFP-Myo2p as a node and contractile ring marker. *∆ain1* cells exhibited clumps of mEGFP-Myo2p (Figure 1A,B) [6]. The clumping phenotype in *∆ain1* began during node coalescence and ended during ring maturation, after the ring is assembled but before the onset of constriction (Figure 1A,C, Appendix A). We calculated that 54% of contractile rings (n = 158 cells) exhibited clumps of mEGFP-Myo2p in *∆ain1* cells (Figure 1D) [6]. The mEGFP-Myo2p clumps in *∆ain1* cells lasted 19.7 ± 7.2 min (mean ± standard deviation, Figure 1E). Forty-eight percent of the rings with clumps of mEGFP-Myo2p exhibited a severe phenotype characterized by a discontinuous fluorescent signal with large gaps in the ring or stretches of intense mEGFP-Myo2p signal separated by dim signal (n = 27 cells; Figure 1A,F). The remaining contractile rings with clumps exhibited mild phenotypes with mild contrast between the stretches of brighter and dimmer fluorescent signal (Figure 1A,F). The clumping phenotype suggests that nodes aggregate into clumps during contractile ring assembly in the absence of Ain1p. This clumping phenotype was previously attributed to the collapse of unstable actin filament structures during node condensation [6]. Therefore, the crosslinking of actin filaments into a network of actin filament bundles contributes to the assembly of a uniform contractile ring.

The clumping phenotype in *∆ain1* cells resolves during the period of contractile ring maturation before the onset of constriction, suggesting that other proteins can compensate for the absence of Ain1p at that time. Therefore, we expected to measure an increase in the duration of clumps in cells with a delayed onset of constriction. Cells that lack Myp2 have a ~14 min delay in the onset of ring constriction [1]. We acquired timelapse micrographs of *∆ain1 ∆myp2* cells expressing mEGFP-Myo2p to measure the duration of clumping when the onset of constriction is delayed. Forty-four percent of *∆ain1 ∆myp2* cells (n = 87 cells) showed the ring clumping phenotype, similar to *∆ain1* cells (Figure 1D). The onset of contractile ring constriction is delayed by ~10 min in *∆ain1 ∆myp2* cells. We measured the duration of clumps in *∆ain1 ∆myp2* cells and found that as expected, clumps last ~7 min longer in those cells (Figure 1E). As measured in *∆ain1* cells, the clumps formed during node coalescence in *∆ain1 ∆myp2* cells but resolved later during the extended phase of maturation, resulting in their longer lifespan. In all but one ring, the clumps resolved before the onset of constriction. The mechanisms that initiate constriction may thus resolve the clumps of nodes.

We acquired timelapse micrographs of *∆ain1 ∆myo51* cells expressing mEGFP-Myo2p and characterized the node clumping phenotype. In this genetic background, 43% of the cells (n = 84 cell) with a newly assembled contractile ring exhibit clumps of mEGFP-Myo2p, comparable to *∆ain1* cells (Figure 1A,B,D). In *∆ain1* cells, clumping began during coalescence and ended before the onset of constriction, lasting 19.7 ± 7.2 min, similar to the *∆ain1 ∆myo51* cells (20.0 ± 9.9 min) (Figure 1C,E, Appendix A). However, in contrast to *∆ain1* cells, *∆ain1 ∆myo51* cells exhibited few contractile rings with severe clumping defects (Figure 1F). Only 12% of contractile rings with clumps showed a severe clumping defect in *∆ain1 ∆myo51* cells (n = 26 cells) compared to 48% in the *∆ain1* cells. To quantify the difference in the severity of the ring clumping phenotype, we measured the fluorescence intensity of the mEGFP-Myo2p in the clumps and normalized it to the fluorescence intensity of the ring (see Section 2 for details). We found that clumps in the *∆ain1* cells were on average 51% brighter than in the double mutant, suggesting that they contained more mEGFP-Myo2p labeled nodes (Figure 1G). Therefore, removing Myo51 in *∆ain1* partially rescues the clumping phenotype by reducing the nodes that accumulate in the clumps. However, the loss of Myo51 in *∆ain1* cells neither prevented the clumping of nodes nor reduced the penetrance of the clumping phenotype across the cell population. The presence of Myo51 in *∆ain1* cells may thus drive more nodes into the clumps. This observation also suggests that Myo51 does not simply function as a crosslinker of actin filaments, as we would have expected of an enhanced clumping phenotype in the *∆ain1 ∆myo51* cells.

Although *∆ain1 ∆myo51* cells exhibit a node clumping defect during contractile ring assembly, no multiseptated or binucleate cells were observed in the population (n = 150 cells) as observed in *∆ain1* and *∆myo51* cells, suggesting that the contractile rings in those cells were still functional. In addition, the duration of maturation and the constriction rate of *∆ain1 ∆myo51* cells is similar to that of the wild-type cell, albeit significantly longer (Appendix A).

### 3.2. The Contractile Ring Sheds Proteins of the Outer Layer during Constriction

During our investigation of the genetic interaction between *ain1* and *myo51*, we noticed that the appearance of the contractile ring during constriction was different in *∆ain1 ∆myo51* cells compared to wild-type, *∆ain1* and *∆myo51* cells. In *∆ain1 ∆myo51* cells expressing mEGFP-Myo2p, bright fragments of the contractile ring separated and extended away from the ring during constriction. Wild-type cells expressing mEGFP-Myo2p also showed these fragments, but the fragments were dimmer and appeared only at the end of constriction (Figure 2A). These fragments of mEGFP-Myo2p align with the ingressing septum in both presumptive daughter cells in the same plane as the constricting contractile ring (Figure 2A). Because these fragments of the ring appear to be leaving the constricting contractile ring at the end of constriction just prior to the disappearance of the contractile ring in wild-type cells, we termed this phenomenon “shedding” (Figure 2A and Figure 3A,E, Appendix A).

We acquired timelapse confocal micrographs of wild-type cells expressing other node proteins labeled with mEGFP and observed a similar shedding phenomenon (Figure 2B). Indeed, other proteins of the outer layer exhibit shedding, including the node proteins mEGFP-Cdc15p, mEGFP-Rng2p and Cdc12p-3GFP (Figure 2B). In addition, mEGFP-Imp2p, another FBAR protein of the contractile ring but not a confirmed component of nodes, also showed shedding (Figure 2B). When the cells were imaged standing upright in yeast motels, specialized microfabricated chambers, the fragments appeared as dynamic filamentous tendrils connected to and extending away from the shrinking ring (Figure 2C). Disconnected fragments also line the ingressing septum, suggesting that fragments shed from the ring until they separate, and the proteins diffuse into the cytoplasm.

We wondered whether shedding was a phenomenon specific to proteins of the outer layer of the ring. We imaged cells expressing mCherry-Myp2p, a component of the inner layer of the contractile ring, by timelapse confocal microscopy. The cells expressing mCherry-Myp2p also showed mCherry-Myp2p extending from the ring, but its appearance was different from that observed with the proteins of the outer ring layer. Near the end of constriction, the mCherry-Myp2p-labeled ring became less uniform with short extensions pointing in any direction with no bias for alignment with the ingressing septum (Figure 2D). Finally, at the end of constriction, mCherry-Myp2p split into two dynamic spots with each daughter cell inheriting one spot. We determined whether Myo51p-3GFP sheds from the contractile ring by imaging cells by timelapse confocal microscopy. As previously reported, we observed that Myo51p-3GFP leaves the constricting contractile ring earlier than other cytokinetic proteins (Figure 2E). Near the end of constriction, Myo51p-3GFP leaves the constricting contractile ring and appears to distribute into small aggregates in the cytoplasm that likely represent Myo51p-3GFP associated with actin filaments. Based on our observations, proteins of the outer layer of the ring shed from the constricting contractile ring late during the constriction phase, while proteins of the inner layers are not detectable in the shedding fragments. This notable difference may be related to the plasma membrane binding affinity of node proteins and Imp2p, whereas the Myp2 and Myo51 molecules have no attachment to the plasma membrane and therefore are not confined to the plane of the plasma membrane.

The timing of the onset of shedding differed according to the cytokinetic protein observed. In wild-type cells expressing mEGFP-Myo2p, shedding begins when the contractile ring is 62% constricted (n = 38 cells), corresponding to an average circumference of 4.2 µm, assuming a full-size ring of 11 µm in circumference. Interestingly, Cdc15p-mCherry appears to shed earlier when the ring is 57% constricted (n = 20 cells), corresponding to a ring of 4.8 µm in circumference. Finally, mCherry-Myp2p extensions appear later than both Myo2p and Cdc15p when the ring is 69% constricted (n = 20 cells), corresponding to a ring of 3.4 µm in circumference. Together, these results suggest that Cdc15p sheds earlier than the Myo2 molecule, suggesting that these two proteins may not be associated into nodes at that time and that the node structure may be dismantling during shedding.

We imaged live fission yeast cells expressing mEos3.2-Rng2p, Myo2p-mEos3.2 or Cdc15p-mEos3.2 by SMLM to determine the organization of these proteins in shedding fragments at the nanoscale. We focused on cells with contractile rings that were very constricted and imaged those cells across their medial plane. SMLM micrographs of very constricted contractile rings show the ring in the center of the cell flanked by shedding fragments extending along the ingressing septum (Figure 2F). At that resolution, the septum is apparent at the center of the shedding fragments. At the nanoscale resolution, Myo2p, Rng2p and Cdc15p mostly appeared as amorphic clouds of proteins along the plasma membrane with some clusters of proteins along the ingressing septum (Figure 2G). This organization is visibly different from the clustered node organization observed in contractile rings that are less than 50% constricted [8,9]. Based on these observations, these proteins may no longer be organized into nodes inside the shedding fragments. Instead, the node organization may be fragmenting during that phase of contractile ring disassembly.

### 3.3. Loss of Both Myo51 and Ain1p Causes Premature and Exaggerated Contractile Ring Shedding

During our investigation, we noticed that contractile ring shedding appeared premature and exaggerated in *∆ain1 ∆myo51* cells when compared to wild-type cells. In wild-type cells expressing mEGFP-Myo2p, shedding begins when the ring is 62% constricted (n = 38 cells), and the fragments that shed from the constricting contractile ring extend out along the septum without reaching the full width of the cell (Figure 2A and Figure 3A–C). In wild-type cells, the contractile ring is easily distinguishable from the flanking shedding fragments as the fluorescence intensity of the fragments is dimmer than that of the contractile ring (Figure 2A,B and Figure 3A,B). We acquired timelapse confocal micrographs of *∆ain1* and *∆myo51* cells expressing mEGFP-Myo2p and quantified their shedding phenotype. In wild-type, *∆ain1* and *∆myo51*, the onset of shedding occurs when the contractile ring is 62% (n = 38 cells), 58% (n = 44 cells) and 65% (n = 42 cells) constricted, respectively (*p* < 0.05 for all pairwise comparison), suggesting that deleting *myo51* causes a slight delay in shedding, while deleting Ain1p causes a slight premature onset in the shedding process (Figure 3C). Although these differences were significantly different, they correspond to small changes in ring circumference. In both *∆ain1* and *∆myo51* single mutants, the duration of shedding was similar to wild-type (Figure 3D). The timing of the onset of shedding compared to SPB separation was greatly different between the different genotypes (Figure 3E, Appendix A). However, this is likely because of the cumulative duration of other cytokinetic events that occur prior to the onset of shedding (compare Figure 1C and Figure 3E, Appendix A). Finally, the fluorescence intensity and length of the shedding fragments flanking the constricting contractile ring were similar between wild-type and *∆myo51* cells (Figure 3A,B,F). In contrast, more rings exhibited exaggerated shedding in *∆ain1* compared to wild type (Figure 3A,B,F).

*∆ain1 ∆myo51* cells expressing mEGFP-Myo2p exhibited exaggerated shedding that began earlier and lasted longer than in the cells carrying a single gene deletion in either *ain1* or *myo51* and wild-type cells (Figure 3A–F, Appendix A). In *∆ain1 ∆myo51* cells, shedding started prematurely when the ring was 51% constricted (n = 51 cells) or an average circumference of 5.4 µm (Figure 3C). This onset of shedding corresponded to 70.2 ± 5.9 min after SPB separation (Figure 3E, Appendix A). In addition, *∆ain1 ∆myo51* cells showed longer and brighter fragments during shedding compared to wild-type, *∆myo51* and *∆ain1* cells (Figure 3A,B,F). In ~70% of the cells, the fragments appeared more continuous with the contractile ring as they extended towards and often reached the lateral edges of the cell, highlighting the entire septa of both daughter cells (Figure 3A,B,F). These fragments were significantly longer in *Δain1 Δmyo51* cells compared to wild-type, *∆ain1* and *∆myo51* cells (Appendix A). In addition, the shedding fragments were often so bright that we could not unambiguously distinguish the constricting contractile ring from the fragments. Taken together, our measurements suggest that the loss of both the Myo51 molecule and Ain1p results in early and exaggerated contractile ring shedding, supporting the cooperation of these two proteins during contractile ring disassembly.

Although *∆ain1 ∆myo51* cells showed exaggerated shedding, these cells had a normal growth, and no multiseptated cells were observed in the population, suggesting that their contractile rings were generally functional and that premature and exaggerated shedding did not interfere with the constriction of the contractile ring (Appendix A).

### 3.4. Myo51 Localizes to the Inner Layer of the Contractile Ring during Constriction

Based on the phenotypes we characterize in this work, the role of Myo51 in cytokinesis likely impacts nodes and their dynamics. Cytokinetic proteins distribute into distinct layers during contractile ring constriction with Myo2-containing nodes in the outer layer closest to the plasma membrane, while Myp2 and the bulk of the network of actin filaments localize to the inner layer, ~100 nm away from the plasma membrane (Figure 4A) [1,10]. This layered distribution is established before the onset of constriction, and Myo51 localizes to the inner layer of the contractile ring prior to the onset of constriction [10]. To confirm that the localization of Myo51p to the inner layer is maintained during constriction, we observed the colocalization of Myo51p with the inner layer protein Myp2 or the outer layer node protein Cdc15p around the ring during constriction. We imaged cells expressing Myo51p-3GFP in combination with either Cdc15p-mCherry or mCherry-Myp2p held upright in yeast motels to visualize the entire ring [1]. We expected Myo51p-3GFP to colocalize with the inner ring molecule Myp2 but not with the outer ring protein Cdc15p. Myo51p-3GFP localized with mCherry-Myp2p in the inner layer but the distribution of the two myosins did not perfectly overlap around the circumference of the ring (Figure 4A). At the onset of constriction, mCherry-Myp2p adopts a crescent shape around the circumference of the ring. This distribution profile remains mostly stable throughout constriction until the ring is smaller and the two ends of the mCherry-Myp2p crescent connect and make a continuous ring. In contrast, Myo51p-3GFP distributes around the circumference of the ring more uniformly with some dynamic brighter puncta, suggesting that Myo51p-3GFP accumulates locally within the inner layer of the ring. Myo51p-3GFP did not colocalize with the node protein Cdc15p-mCherry around the ring during constriction. Instead, Cdc15p-mCherry localized to the outer layer closest to the plasma membrane consistent with its plasma membrane binding and its localization to membrane-bound cytokinetic nodes, while the Myo51p-3GFP signal was detectable deeper in the cell (Figure 4A). Peels of Myo51p-3GFP occasionally separated from the main ring and moved centripetally inside the constricting contractile ring (Figure 4A). Together these data support the localization of Myo51p-3GFP to the inner layer of the contractile ring, where mEGFP-Myp2p and the bulk of the actin filament network localize during constriction.

These data and previously published super-resolution data support the localization of Myo51p-3GFP to the inner layer of the constricting contractile ring where the bulk of the actin filament bundle resides [10]. These results are also consistent with the dependence of Myo51 on the presence of actin filaments for its localization to the contractile ring [25]. When actin is depolymerized by Latrunculin A (LatA), the contractile ring separates into individual nodes, suggesting that actin filaments connect and maintain the nodes into a ring [8,12]. We determined whether Myo51p-3GFP dispersed with Cdc15p-mCherry nodes or whether Myo51p-3GFP diffused into the cytoplasm upon LatA treatment. To visualize Myo51-3GFP leaving the contractile ring, we had to use a very low dose of LatA, 2.5 µM. We treated cells expressing Cdc15p-mCherry and Myo51p-3GFP with DMSO or 2.5 µM LatA, rapidly prepared cells for imaging and acquired timelapse micrographs by confocal microscopy within ~6 min of the addition of the drug or solvent. The Cdc15p-mCherry and Myo51p-3GFP signals in the DMSO-treated cells remained confined to the contractile ring throughout the 60 min imaging session (Figure 4B). In cells treated with 2.5 µM LatA, the actin filament network slowly and partially broke down. Even with a partial breakdown of the actin filament network, the Cdc15p-mCherry rings already appeared discontinuous at the start of imaging (~6 min after the addition of LatA) and Cdc15p-mCherry labeled nodes started to disperse away from the plane of division as previously observed, suggesting that the actin filaments that maintain the nodes into a ring are disassembling [8,12]. At the start of imaging, many cells had already lost most of their Myo51p-3GFP signal from their ring. In the cells that still had a dim signal, puncta of Myo51p-3GFP aligned with the position of the disassembling ring (Figure 4B). These puncta did not disperse away from the plane of the ring like the Cdc15p-mCherry nodes. Instead, the fluorescence intensity of the puncta dimmed until they were no longer visible. We measured the global fluorescence of Myo51p-3GFP in whole cells in DMSO- or LatA-treated cells and found no significant difference in the total amount of Myo51p-3GFP, suggesting that Myo51p-3GFP leaves the disassembling contractile ring and diffuses into the cytoplasm (Appendix A). Together, our data and other published work suggest that Myo51 localizes to the inner ring of the contractile ring to the main bundle of actin filaments during constriction in an actin-dependent manner and suggest that the primary functions of Myo51p during cytokinesis are related to the actin network of the contractile ring. Therefore, Myo51 may impact node dynamics indirectly by acting in the main bundle of actin filaments during cytokinesis.

## 4. Discussion

In this work, we investigated the potential function of Myo51 as a crosslinker of actin filaments. We hypothesized that removing Myo51 in cells that lack the passive actin filament crosslinker Ain1p (*∆ain1* cells) would enhance the *∆ain1* node clumping phenotype observed during contractile ring assembly [6]. Unexpectedly, removing Myo51 from *∆ain1* partially rescued the node clumping phenotype observed in single mutant *∆ain1* cells. If Myo51 does not act as an actin filament crosslinker, how does it support cytokinesis? During contractile ring assembly, loss of Myo51 results in an increase in the population of immobile nodes, suggesting that Myo51 may help nodes engage with the actin filament network, allowing them to move [2].

Clumps of Myo2 likely represent the non-uniform distribution of nodes within the contractile ring. Different factors can lead to the formation of clumps of nodes. Rings with clumps of nodes occur in cells with defective “release” in the SCPR mechanism, as seen in cells with mutations in the actin filament severing factor cofilin/Adf1p and in simulations of the SCPR mechanism [4,5]. The loss of the actin filament crosslinker Ain1p also results in the clumping of nodes during contractile ring assembly [6]. Simulations of the minimal SCPR mechanism recapitulate the assembly of the contractile ring by the coalescence of a band of nodes without the need for actin filament crosslinkers. Although the SCPR mechanism may be sufficient to coalesce nodes, the addition of “node alignment” onto bundles of actin filaments made the SCPR mechanism more robust [6,7]. Indeed, SMLM of cells in the ring assembly phase show nodes aligned onto linear structures that likely represent actin bundles. Although we observe this phenomenon in SMLM of wild-type cells, the linear structures are more obvious in *cdc25-22* cells arrested and released, likely because the cells are longer and assemble a wider band of nodes [8,9]. The increased distance across the band of nodes may facilitate the observation of these structures. The node clumping defect in *adf1* mutant cells is exacerbated by the additional loss of the *ain1* gene, suggesting that the release of actin filament connections between nodes and the alignment of nodes onto actin bundles both contribute to the assembly of a contractile ring with a uniform distribution of nodes [5].

According to the updated SCPR mechanism, actin filament crosslinkers such as Ain1p bundle actin filaments may increase the overall continuity across the actin filament network that connect the nodes (Figure 5A). Therefore, the lack of Ain1p likely reduces the connectivity across the network of nodes, resulting in a greater porosity and clumps of nodes [6,7]. Myo51 supports node motions promoting their coalescence into the assembling ring [1,2]. Efficient node motions in a poorly connected actin filament network in *∆ain1* cells possibly help the formation of clumps. Alternatively, removing Myo51 likely impedes node motions, resulting in clumps with fewer nodes, as we observed in the partial rescue of the clumping phenotype in *∆ain1 ∆myo51* cells.

The clumping phenotype of *∆ain1* and *∆ain1 ∆myo51* cells resolves during ring maturation ahead of constriction without delaying the onset of constriction, suggesting that the recruitment of factors during maturation of the ring may redistribute the protein complexes in the outer ring layer in these mutants. Consistent with this interpretation, deletion of *myp2* delays the resolution of clumps, supporting that Myp2 may contribute to this mechanism. The production of contractile forces ahead of constriction may also help redistribute the nodes within the contractile ring.

How the contractile ring disassembles while producing force is one of the major remaining questions about cytokinesis [26]. In this work, we describe the phenomenon of shedding, a normal process by which proteins appear to exit the contractile ring during the second half of constriction. Every protein from the outer layer of the contractile ring that we observed in this work sheds, and their shedding begins when the ring is more than 50% constricted. During shedding, cytokinetic proteins from the outer layer of the ring exit the contractile ring and align along the ingressing septum in both presumptive daughter cells. The SMLM of contractile shedding suggested that proteins in the shedding fragments may not be assembled into organized node structures. Indeed, the observation that Cdc15p sheds earlier than Myo2p supports that the nodes may be dismantling in the shedding fragments. The proteins of the outer ring layer may be shedding along the ingressing septum because these proteins are directly or indirectly associated with the plasma membrane restricting their distribution. During shedding, proteins of the outer layer of the contractile ring are sometimes organized into filamentous fragments, suggesting that the proteins may be connected to a bundle of actin filaments. The network of actin filaments of the contractile ring at this stage of constriction is obscured by the dynamic actin patches that assemble along the ingressing septum. We therefore could not unambiguously identify bundles of actin filaments among the brighter actin patches in constricted contractile rings. Why cytokinetic proteins shed from the constricting contractile ring near the end of constriction remains to be determined. It is conceivable that the structural organization of the contractile ring changes as its circumference decreases. The curvature of the ring may force this change in organization, resulting in the onset of protein shedding.

Proteins of the inner layer of the contractile ring must also leave the ring but appear to do so differently from the shedding observed for the proteins of the outer layer. When the ring is ~70% constricted (ring of ~3 µm in circumference), short and thick projections of mCherry-Myp2p signal extend from the ring in any direction. As Myp2 associates with the bundle of actin filaments in the inner layer of the contractile ring and not with the plasma membrane, the appearance of mCherry-Myp2p may therefore highlight the breaking down of the underlying bundle of actin filaments during the late stage of constriction. Myo51p-3GFP leaves the ring and appears in the cytoplasm as small aggregates, likely representing groups of Myo51p-3GFP on actin filament bundles.

Our work shows that the combined loss of Myo51 and Ain1p results in premature and exaggerated contractile ring shedding. The loss of actin filament crosslinking by Ain1p may sensitize the contractile ring to other perturbations that can lead to an exaggerated ring shedding phenotype, such as the loss of Myo51 (Figure 5B). The function of Myo51 during ring shedding is harder to interpret. Unlike during ring assembly, Myo51 may function as a crosslinker of actin filaments during constriction. The combined loss of two actin filament crosslinkers may destabilize the actin network and thus lead to premature and exaggerated shedding with an elongated shedding period. Alternatively, Myo51 may maintain the same role during contractile ring assembly, constriction and shedding: to engage nodes with the actin filament network. Partial loss of engagements between nodes and the actin filament network, combined with the loss of Ain1p crosslinking of actin filaments may result in premature and exaggerated shedding.

The loss of Ain1p from the contractile ring results in the ectopic localization of fimbrin/Fim1p to the contractile ring as the two passive crosslinkers compete for actin filament binding [18,19]. The presence, the concentration and the timing of recruitment of Fim1p to the contractile ring of *∆ain1 ∆myo51* cells may help reconcile the complex interplay between Myo51 and Ain1p during cytokinesis. Future work will be necessary to determine whether and how Fim1p may be involved in the clumping and shedding seen in *∆ain1 ∆myo51* cells.

## Figures and Tables

**Figure 1 jof-10-00647-f001:**
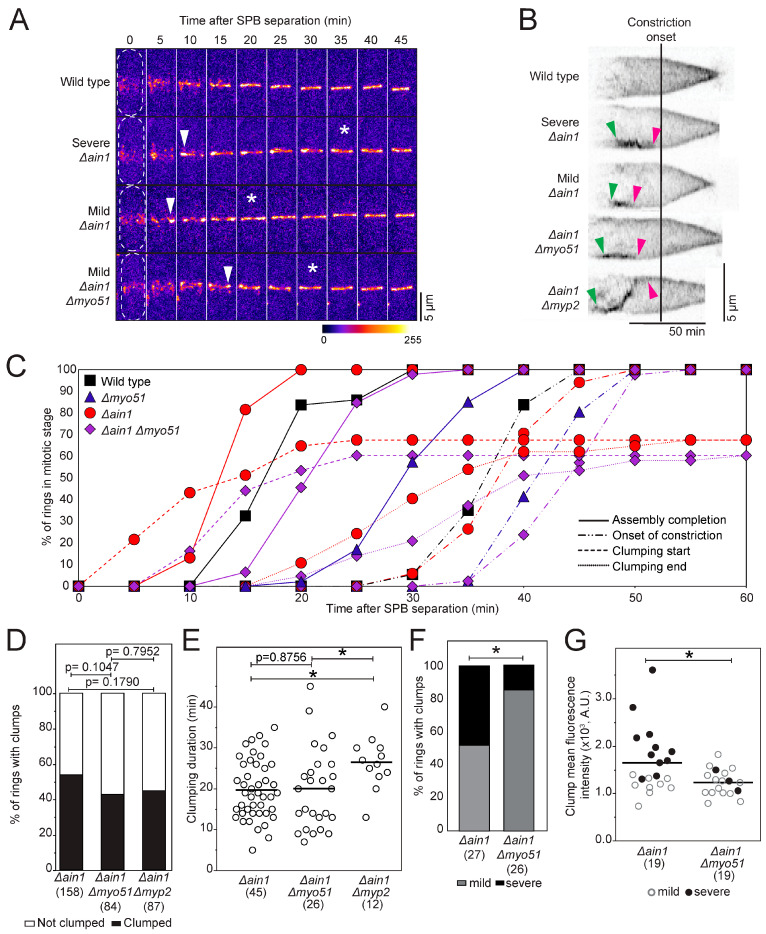
*Δmyo51 Δain1* cells partially rescue node clumping phenotype seen in *∆ain1* cells. (**A**) Confocal micrograph (Fire LUT) of contractile rings in cells expressing mEGFP-Myo2p. Arrowheads, clumping starts. Asterisks, clumping ends. Dashed lines, cell outlines. (**B**) Representative kymographs aligned at constriction onset (inverted grayscale LUT) of constricting contractile rings from cells expressing mEGFP-Myo2p. Green arrowheads, clumping starts. Magenta arrowheads, clumping ends. (**C**) Outcomes plot of the timing of assembly completion (n = 38–46), onset of constriction (n = 34–42), clumping start (n = 37–43) and clumping end (n = 37–43) in cells expressing mEGFP-Myo2p. Significance determined by log rank Test and reported in Appendix A. (**D**) Stacked bar chart of the percentage of rings that display mEGFP-Myo2p clumping phenotype. Significance determined by Pearson’s chi-square test. (**E**) Swarm plot of the duration of clumping for cells expressing mEGFP-Myo2p. Bars, means. Asterisks, *p* < 0.05 by 2-tailed Student’s *t* test. (**F**) Stacked bar chart of the percentage of cells expressing mEGFP-Myo2p that display mild (23% of *Δain1* cells, 38% of *Δain1 Δmyo51* cells among whole population) or severe (29% of *Δain1* cells, 5% of *Δain1 Δmyo51* cells among whole population) clumping phenotype. Asterisk, *p* < 0.05 by Pearson’s chi-square test. (**G**) Swarm plot of the mean fluorescence intensity of the clumps for cells expressing mEGFP-Myo2p. Bars, means. Asterisk, *p* < 0.05 by 2-tailed Student’s *t* test.

**Figure 2 jof-10-00647-f002:**
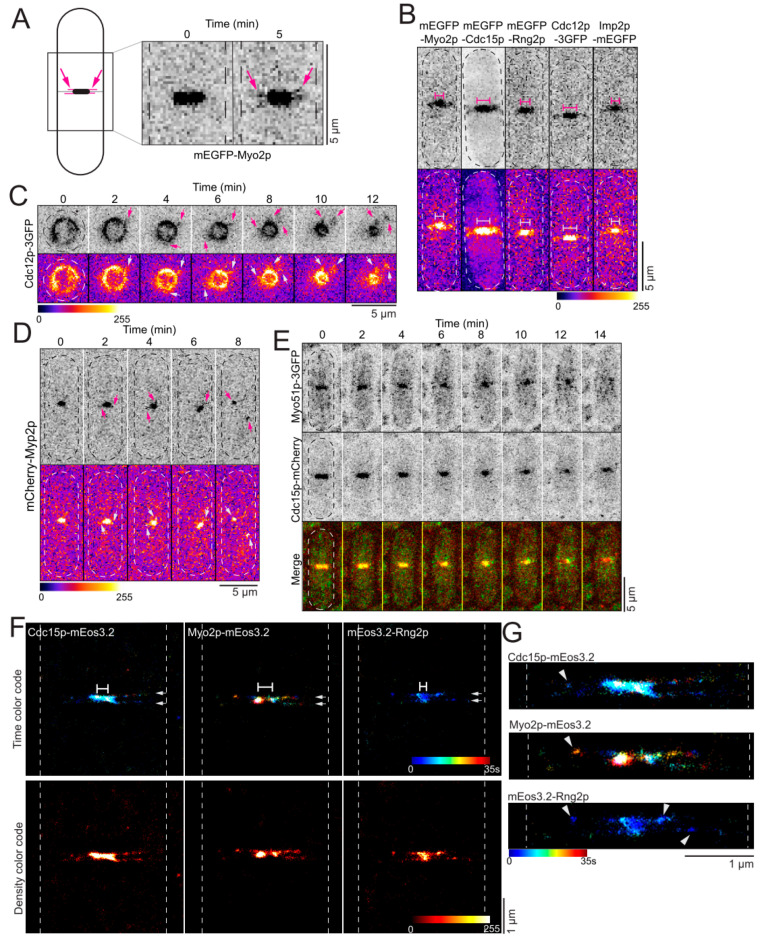
Outer ring proteins “shed” from contractile ring in wild-type cells. (**A**) Model of a shedding ring (left). Inset confocal micrograph (inverted grayscale LUT) of a representative wild-type cell expressing mEGFP-Myo2p undergoing shedding (right). Magenta arrows, shedding fragments. Dashed lines, cell outlines. (**B**) Confocal micrograph (inverted grayscale LUT, top and Fire LUT, bottom) of cells expressing different fluorescently labeled proteins. Brackets, contractile rings. Dashed lines, cell outlines. (**C**) Timelapse confocal micrograph (inverted grayscale LUT, top and Fire LUT, bottom) of a cell expressing Cdc12p-3GFP imaged in a yeast motel. Arrows, shedding fragments. Dashed lines, cell outlines. (**D**) Timelapse confocal micrograph (inverted grayscale LUT, top and Fire LUT, bottom) of a cell expressing mCherry-Myp2p. Arrows, shedding fragments. Dashed lines, cell outlines. (**E**) Timelapse confocal micrograph of a cell (inverted grayscale LUT and colored merged) expressing Myo51p-3GFP and Cdc15p-mCherry. Dashed lines, cell outlines. (**F**) Representative SMLM images of shedding in wild-type cells color-coded for time (top, Jet LUT) or density (bottom, Hot LUT). Brackets, contractile rings. Arrows, shedding material along septum. Dashed lines, cell outlines. (**G**) Enlarged SMLM images of shedding in wild-type cells from (**F**), color-coded for time. Arrowheads, shedding material. Dashed line, cell outline.

**Figure 3 jof-10-00647-f003:**
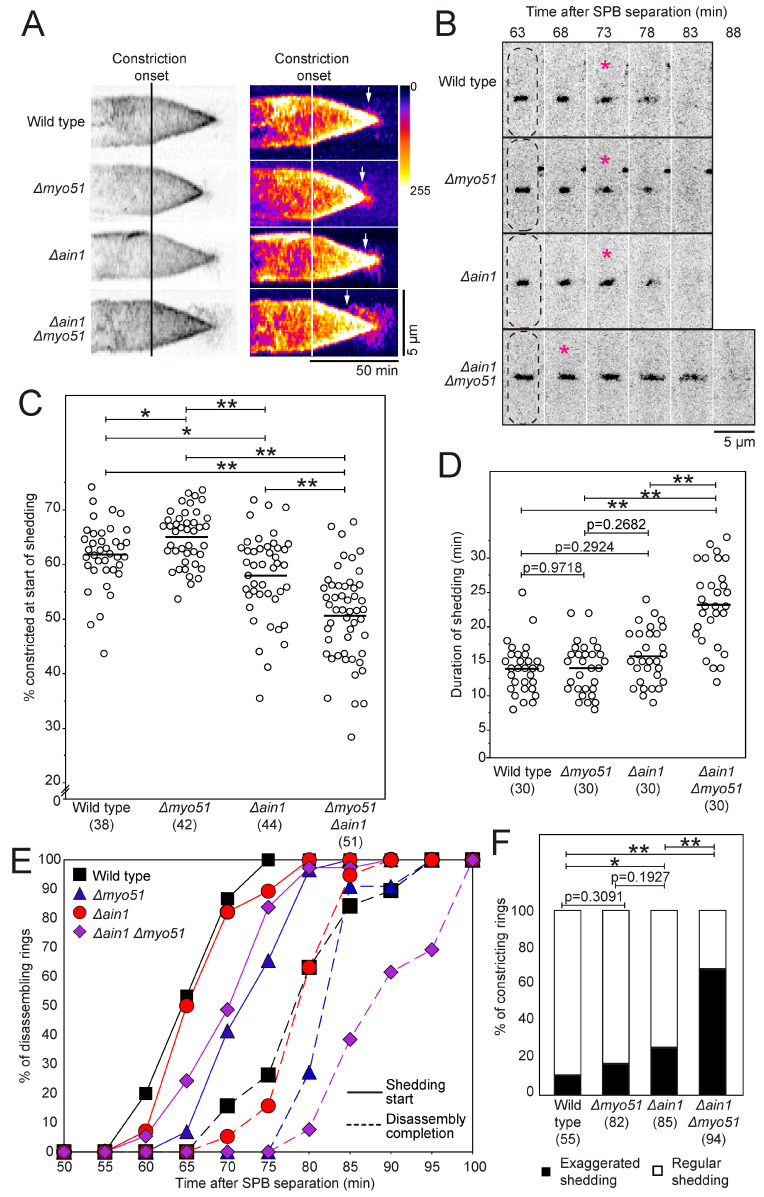
Contractile ring shedding is exaggerated and premature in *Δain1 Δmyo51* cells. (**A**) Representative kymographs of contractile rings aligned at constriction onset from cells expressing mEGFP-Myo2p (inverted grayscale LUT, left and Fire LUT, right). Arrows, shedding starts. (**B**) Timelapse of confocal micrographs (inverted grayscale LUT) for cells expressing mEGFP-Myo2p. Magenta asterisks, first depicted timepoint with visible shedding. Dashed lines, cell outlines. (**C**) Swarm plot for the percentage of contractile ring constriction when shedding begins for cells expressing mEGFP-Myo2p. Bars, means. Single asterisks, *p* < 0.05 by 2-tailed Student’s *t* test. Double asterisks, *p* < 0.0001 by 2-tailed Student’s *t* test. (**D**) Swarm plot of the duration of shedding for cells expressing mEGFP-Myo2p. Bars, means. Double asterisks, *p* < 0.0001 by 2-tailed Student’s *t* test. (**E**) Outcomes plot showing the timing of the onset of shedding (n = 39–54) and the completion of ring disassembly (n = 30) in cells expressing mEGFP-Myo2p. Significance determined by log rank test and reported in Appendix A. (**F**) Stacked bar chart of the percentage of cells expressing mEGFP-Myo2p that display an exaggerated shedding phenotype. Single asterisk, *p* < 0.05 by Pearson’s chi-square test. Double asterisks, *p* < 0.0001 by Pearson’s chi-square test.

**Figure 4 jof-10-00647-f004:**
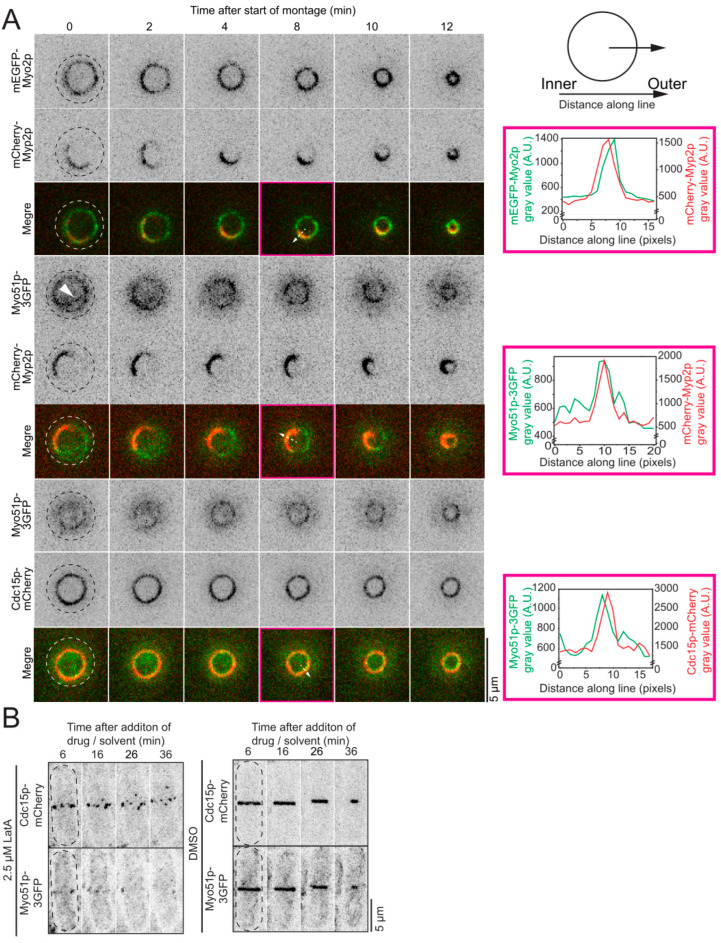
Myo51p localization to the inner layer of the contractile ring is actin dependent. (**A**) Timelapse confocal micrographs (inverted grayscale LUT and colored merged images) of cells imaged in a yeast motel expressing different pairs of fluorescently labeled cytokinetic markers. Model of orientation and method of line scan (top right). Graphs (right) depicting fluorescence intensity for both fluorescent signals along the line for each individual micrograph. Dashed lines, cell outlines. Magenta boxes, time frames used for line scans. Dashed arrows, lines used for fluorescence intensity scan. Arrowhead points to Myo51p-3GFP peel. (**B**) Timelapse confocal micrograph (inverted grayscale LUT) of 2.5 µM LatA treated (left) and DMSO treated (right) cells. Dashed lines, cell outlines.

**Figure 5 jof-10-00647-f005:**
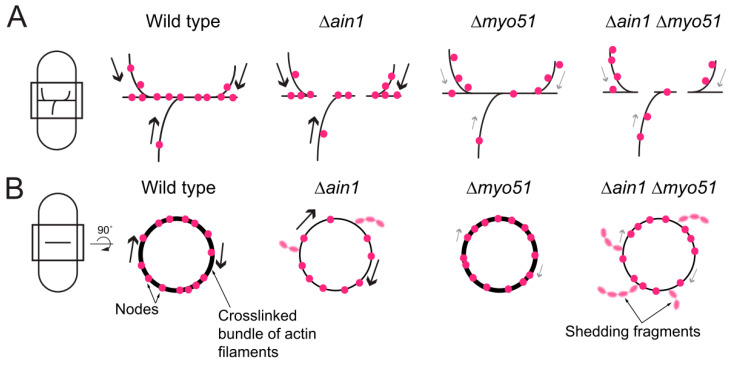
Model of clumping and shedding in the contractile ring. (**A**) Model of assembling contractile ring within a fission yeast cell (left). Models of assembling contractile rings to demonstrate clustering of nodes in wild-type, *Δmyo51*, *Δain1* and *Δmyo51 Δain1* cells (right). Magenta circles, nodes. Arrows, node movements. (**B**) Model of constricting contractile ring within a fission yeast cell (left). Models of constricting contractile rings in upright orientation to demonstrate shedding of material in wild-type, *Δmyo51*, *Δain1* and *Δmyo51 Δain1* cells (right). Magenta circles, nodes. Dim magenta ovals, node material in shedding fragments. Arrows, node movements.

## Data Availability

The original contributions presented in the study are included in the article/Appendix A, further inquiries can be directed to the corresponding author.

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
