# Peer review of "The Myosin-V Myo51 and Alpha-Actinin Ain1p Cooperate during Contractile Ring Assembly and Disassembly in Fission Yeast Cytokinesis"

_jof, 2024, doi:10.3390/jof10090647_

Round 1

Reviewer 1 Report

In this study, Tyree et al hypothesized that Myo51, a myosin with the capacity to bind actin filaments at both its motor domain and tail, may act as a crosslinker and collaborate with the passive actin filament crosslinker Ain1 during cytokinesis. Their findings led them to propose that Myo51 does not solely operate as a crosslinker, as the absence of Myo51 did not exacerbate the phenotype of cells lacking Ain1. To better understand the functional role of Myo51 in the cytokinetic process, they authors conducted a detailed analysis of the localisation of cytokinetic node proteins during the assembly and constriction stages in the absence of Ain1, Myo51, and both. During their investigation, they discovered a new phenomenon called “shedding”, in which fragments of the contractile ring disperse away from the ring. Based on their data, the authors suggested that during assembly, Myo51 may favour the movement of nodes, whereas during constriction, it may act as a crosslinker.

The researchers carried out, presented, and included, mostly, all relevant controls in their experiments. However, the results were descriptive in nature, and the biological implications of the phenotypes observed were not examined in detail. Moreover, there might be some extrapolation of the results obtained.

Major points:

Figure 1. Panel B of Figure 1 reveals a decrease in Myo51-3GFP intensity within the ring, which could be attributed to either protein delocalization or degradation. To provide further insight, it is recommended that authors include an analysis of Myo51-3GFP levels using western blot in DMSO and LatA-treated cells as a control. Additionally, the current state of the actin cytoskeleton in cells treated with LatA at the specified dose should be examined and presented, although this information is already available in the literature.

As depicted in Panel A of Figure 1, it is evident that the localization of Myo51 exhibits greater similarity to Myp2 than to other membrane-associated proteins, such as Cdc15, despite the presence of a substantial amount of protein that colocalizes with these proteins. This observation should be emphasized in the text.

Figure 2. In this figure, the authors investigate whether the lack of Ain1, Myo51, or both affects the localization of node protein Myo2. Do the clumps observed in the absence of Ain1, Myo51, or both also appear in other node contractile ring proteins such as Rlc1, Rng2, Cdc12, Imp2, or Cdc15?

Figure 4. In Figure 4, Panel E, it would be advantageous to utilize a different type of graphical representation that would facilitate the visualization of individual values for each cell, as well as an improved visualization of the extent of the delay in cytokinesis dynamics among the various mutants.

Author Response

In this study, Tyree et al hypothesized that Myo51, a myosin with the capacity to bind actin filaments at both its motor domain and tail, may act as a crosslinker and collaborate with the passive actin filament crosslinker Ain1 during cytokinesis. Their findings led them to propose that Myo51 does not solely operate as a crosslinker, as the absence of Myo51 did not exacerbate the phenotype of cells lacking Ain1. To better understand the functional role of Myo51 in the cytokinetic process, they authors conducted a detailed analysis of the localisation of cytokinetic node proteins during the assembly and constriction stages in the absence of Ain1, Myo51, and both. During their investigation, they discovered a new phenomenon called “shedding”, in which fragments of the contractile ring disperse away from the ring. Based on their data, the authors suggested that during assembly, Myo51 may favour the movement of nodes, whereas during constriction, it may act as a crosslinker.

The researchers carried out, presented, and included, mostly, all relevant controls in their experiments. However, the results were descriptive in nature, and the biological implications of the phenotypes observed were not examined in detail. Moreover, there might be some extrapolation of the results obtained.

We thank the reviewer for their thorough readthrough of our manuscript and for their thoughtful comments. We address each concern below. Based on the combined comments from multiple reviewers, we opted to change the order of the text. The section entitled Myo51 localizes to the inner layer of the contractile ring during constriction” can now be found at the end of the manuscript. Consequently, the order of the figures has been updated.

Major points:

Q1: Figure 1. Panel B of Figure 1 reveals a decrease in Myo51-3GFP intensity within the ring, which could be attributed to either protein delocalization or degradation. To provide further insight, it is recommended that authors include an analysis of Myo51-3GFP levels using western blot in DMSO and LatA-treated cells as a control. Additionally, the current state of the actin cytoskeleton in cells treated with LatA at the specified dose should be examined and presented, although this information is already available in the literature.

Answer:

Old Figure 1 is now Figure 4.

To determine whether Myo51p-3GFP degrades upon LatA treatment, we compared the global cytoplasmic fluorescence intensity for DMSO and LatA treated cells to compare the amount of Myo51p-3GFP present after each treatment. We measured no significant difference between these two treatments by a 2-tailed Student’s t-test suggesting that Myo51p-3GFP disperses into the cytoplasm rather than being degraded. Nevertheless, if Myo51p were to be degraded in the cytoplasm in the presence of LatA, this would not affect our result that the localization of Myo51p to the contractile ring during constriction depends on the presence of actin filaments.

We modified the text to clearly state that at such low concentration of LatA, the actin filament network partially breaks down. This partial depolymerization is sufficient to release the Cdc15p-mCherry nodes and cause the dispersal of Myo51p-3GFP.

To address these comments, we updated the section of the Results entitled “Myo51 localizes to the inner layer of the contractile ring during constriction”, which is now located at the end of the manuscript rather than at the beginning. We also included a new Supplemental Figure 2 and figure legend showing the global cytoplasmic fluorescence intensity of Myo51p-3GFP in DMSO and LatA treated cells.

Q2: As depicted in Panel A of Figure 1, it is evident that the localization of Myo51 exhibits greater similarity to Myp2 than to other membrane-associated proteins, such as Cdc15, despite the presence of a substantial amount of protein that colocalizes with these proteins. This observation should be emphasized in the text.

Answer:

Old Figure 1 is now Figure 4.

We have clarified this point with the following changes to the text (new text bolded):

“Peels of Myo51p-3GFP occasionally separated from the main ring and moved centripetally inside the constricting contractile ring (Figure 4A). Together these data support the localization of Myo51p-3GFP to the inner layer of the contractile ring where mEGFP-Myp2p and the bulk of the actin filament network localize during constriction.

Q3: Figure 2. In this figure, the authors investigate whether the lack of Ain1, Myo51, or both affects the localization of node protein Myo2. Do the clumps observed in the absence of Ain1, Myo51, or both also appear in other node contractile ring proteins such as Rlc1, Rng2, Cdc12, Imp2, or Cdc15?

Answer:

Old Figure 2 is now Figure 1.

In this study we used mEGFP-Myo2p as a marker for nodes. We did not determine if there was clumping in other node proteins such as Rng2p, Cdc12p, or Cdc15p as we assume that other node proteins would colocalize with Myo2 based on Laporte et al. 2012. In their work, Laporte et al. 2012 reported clumps of the node proteins Rlc1p, Myo2p, Mid1p and Cdc15 in ∆ain1 cells. We have clarified this information in the text (new text bold).

“Myo51 can bind actin filaments at both its motor domain and tail suggesting that it may act as a crosslinker [3]. Myo51 may therefore cooperate with the passive actin filament crosslinker Ain1p during cytokinesis. It was previously shown by Laporte et al. that ∆ain1 cells expressing other node markers Rlc1p, Cdc15p, Myo2p and Mid1p exhibit clumping [6].

Q4: Figure 4. In Figure 4, Panel E, it would be advantageous to utilize a different type of graphical representation that would facilitate the visualization of individual values for each cell, as well as an improved visualization of the extent of the delay in cytokinesis dynamics among the various mutants.

Answer:

Old Figure 4 is now Figure 3. Old Figure 2 is now Figure 1.

All timing events shown in outcomes plots in new Figures 1C and 3E are now also represented by swarm plots in the new Supplemental Figures 1A-F. We have properly referenced these new figure panels in the text.

Reviewer 2 Report

Myo51 has been suggested to act as an actin crosslinker in fission yeast cytokinesis. Since the deletion of ain1, encoding α-actinin which is another actin crosslinker, results in the node clumping phenotype, the authors examined whether Δmyo51Δain1 enhanced the defect of Δain1, and found that the double knockout rather reduced the severity of the clumping, against their expectation. They therefore conclude that myo51 has other functions than actin crosslinking in cytokinesis.

I evaluate that this manuscript is too premature to be published. The title and introduction are inappropriate, because the present study does not focus on the actin-crosslinking activities of Myo51 and Ain1 and their cooperation, but more extensively investigates the contractile ring constriction. Nevertheless, it failed to contribute to adequate advancement of understanding the mechanisms of cytokinesis. The manuscript is overall highly descriptive and poorly organized. Furthermore, many of the results are mere repetition of previous studies. I describe below the individual comments.

Section 3.1

Considering the aim of this study, I do not understand why the authors started this work with the localizations of these proteins. Furthermore, many of the results and the conclusion described in line 235 – 237 seem to be already reported. Are the crescent localization pattern of Myp2 and the slightly different localizations between the two myosins novel? What do the data represent? Do they contribute to the conclusion in this section? 

In Figure 1B, why do the authors compare the behaviors of Cdc15p and Myo51? What does the difference represent? Don’t the difference result from their different signal intensities?

Section 3.2

In Figure 2, the severe and mild clumping phenotypes should be objectively determined. How were they judged?

Section 3.3

Line 320 – 322

Which data lead to this description?

Line 357 – 359

What do the spots and their behaviors represent?

Section 3.4

‘Exaggerated shedding’ should be objectively determined. State the method to judge it.

I included the comments in 'Major comments'.

Author Response

Myo51 has been suggested to act as an actin crosslinker in fission yeast cytokinesis. Since the deletion of ain1, encoding α-actinin which is another actin crosslinker, results in the node clumping phenotype, the authors examined whether Δmyo51Δain1 enhanced the defect of Δain1, and found that the double knockout rather reduced the severity of the clumping, against their expectation. They therefore conclude that myo51 has other functions than actin crosslinking in cytokinesis.

I evaluate that this manuscript is too premature to be published. The title and introduction are inappropriate, because the present study does not focus on the actin-crosslinking activities of Myo51 and Ain1 and their cooperation, but more extensively investigates the contractile ring constriction. Nevertheless, it failed to contribute to adequate advancement of understanding the mechanisms of cytokinesis. The manuscript is overall highly descriptive and poorly organized. Furthermore, many of the results are mere repetition of previous studies. I describe below the individual comments.

We thank the reviewer for their thorough readthrough of our manuscript and for their thoughtful comments. We address each concern below. Based on the combined comments from multiple reviewers, we opted to change the order of the text. The section entitled Myo51 localizes to the inner layer of the contractile ring during constriction” can now be found at the end of the manuscript. Consequently, the order of the figures has been updated.

Q1: Considering the aim of this study, I do not understand why the authors started this work with the localizations of these proteins. Furthermore, many of the results and the conclusion described in line 235 – 237 seem to be already reported. Are the crescent localization pattern of Myp2 and the slightly different localizations between the two myosins novel? What do the data represent? Do they contribute to the conclusion in this section?

Answer:

In this work, we investigated the potential function of Myo51 as a crosslinker of actin filaments during the constriction of the cytokinetic contractile ring. Before the onset of constriction of the contractile ring, during the maturation phase, Myo51 localizes deep into the assembled contractile ring (McDonald et al.). We set out to confirm that this localization is maintained during constriction using confocal microscopy of cells held vertically in specialized cell holders. We have updated the section entitled “Myo51 localizes to the inner layer of the contractile ring during constriction” to clarify our rationale and moved it to the end of the Result section.

Q2: In Figure 1B, why do the authors compare the behaviors of Cdc15p and Myo51? What does the difference represent? Don’t the difference result from their different signal intensities?

Answer:

Old Figure 1 is now Figure 4.

The difference between the distribution of the Cdc15p-mCherry labeled nodes and the Myo51p-3GFP signal is unrelated to their respective intensities. To clarify this confusion, we updated the text of the section entitled “Myo51 localizes to the inner layer of the contractile ring during constriction”.

Q3: In Figure 2, the severe and mild clumping phenotypes should be objectively determined. How were they judged?

Answer:

Old Figure 2 is now Figure 1.

We included a description of severe and mild clumping in the Methods section in addition to the definition already present in the result section. The text below has been added to the Methods section.

“Classification of Clumping and Shedding Phenotypes

A ring with mild clumping refers to a contractile ring with no gaps in signal but with localized bright and dim spots around the ring. A ring with severe clumping was defined by a discontinuous fluorescent signal with large gaps in the ring or stretches of intense mEGFP-Myo2p signal separated by dim signal. A ring with a moderate clumping phenotype exhibited mild contrast between stretches of brighter and dimmer fluorescent signal.

Exaggerated shedding was defined as shedding fragments that are long enough to reach the width of the cell and bright enough to obscure the body of contractile ring, making it difficult to distinguish the contractile ring from the shedding fragment.”

Line 320 – 322

Q4: Which data lead to this description?

Answer:

Old Figure 3 is now Figure 2.

We ensured that the data illustrating the appearance of shedding in Figures 3A and 4E were properly referenced in the text as follows (new text bolded):

“Because these fragments of the ring appear to be leaving the constricting contractile ring at the end of constriction just prior to the disappearance of the contractile ring in wild-type cells, we termed this phenomenon “shedding” (Figures 2A, 3A and E, Supplemental Figure1E and F).”

Line 357 – 359

Q5: What do the spots and their behaviors represent?

Answer: It is not yet known why Myp2p separates into these two spots and what other proteins may be found in these spots. The sentence referenced here is a description of the distribution of the mCherry-Myp2p signal during cytokinesis.

Q6: ‘Exaggerated shedding’ should be objectively determined. State the method to judge it.

Answer:  We included a description of exaggerated and mild shedding to the Methods section, in addition to the definition already present in the result section. The text below has been added to the Methods section.

“Classification of Clumping and Shedding Phenotypes

A ring with mild clumping refers to a contractile ring with no gaps in signal but with localized bright and dim spots around the ring. A ring with severe clumping was defined by a discontinuous fluorescent signal with large gaps in the ring or stretches of intense mEGFP-Myo2p signal separated by dim signal. A ring with a moderate clumping phenotype exhibited mild contrast between stretches of brighter and dimmer fluorescent signal.

Exaggerated shedding was defined as shedding fragments that are long enough to reach the width of the cell and bright enough to obscure the body of contractile ring, making it difficult to distinguish the contractile ring from the shedding fragment.”

Reviewer 3 Report

This manuscript by Tyree et al. studied the role of type V myosin Myo51 in cytokinesis using fission yeast as the model organism. By characterizing actinin Ain1 and Myo51 deletion mutants, the study demonstrated that contradictory to the authors’ hypothesis deletion of Myo51 results in more stable actin filament structures in the ain1 deletion mutant cells.

The paper describes an interesting interplay between a motor protein and a actin-crosslinker during cytokinesis. The discovery is very surprising. It uses a varies of unique approaches including “yeast motel” and super-resolution microscopy. The results shall be of interest to the field of cytokinesis. However, the reviewer has two major concerns.

1.      The genetic interaction between myo51 and ain1 mutants can be better characterized. For example, does the double mutant grow fast than the single ain1 mutant? Does the number of bi-nuclei cells among the myo51 ain1 mutant decrease, compared to the either myo51 or ain1 mutant? What about the septation indices of the single and double mutants? Lastly, how are the contractile assembly, maturation and constriction of the double myo51 ain1 mutant compared to the single ain1 mutant? Fig. 1C appear to address this last question partially, but this panel is hard to interpret in its current format and without the statistics. All the answers will clarify the genetic interaction between myo51 and ain1.

2.       The phenotype of “shedding” can be characterized more quantitatively. It remains unclear how many fragments are being pushed out of the ring. The duration and size of the shredding shall also be quantified. Lastly, is there a correlation between shedding and the outcome of the contractile ring constriction?

Minor concerns,

1.      Author affiliations are not clear. Missing superscripts.

3.      Typo: Line 34. Shall be cam1p.

4.      Does myo51 deletion alter the localization of the crosslinking protein Ain1 or Fim1? This may provide another potential explanation for the partial rescue of ain1 deletion mutant by the myo51 mutant.

5.      Fig. 1B What is the evidence that the actin cytoskeletal structures is disrupted in the presence of LatA for 6 mins?

6.      Fig. 1A: statistics of the distance between inner and outer contractile ring.

7.      Fig. 2A: please indicated percentage of the phenotypes (sever, mild) among the mutants

8.      Fig. 2B: the ring in the top panel constricted earlier than the vertical line indicates.

9.      Fig. 2C is very confusing. Please consider to split into multiple panels.

10. Fig. 2E-G: missing standard deviations

11. Fig. 3: please add the LUT for fire pseudo-colored micrographs.

12. Fig. 4A: Missing LUT of the pseudo color. Please clarify when the “shedding” starts.

13. Fig. 4C is confusing. Does the Y axis mean that the gap between the start of the ring constriction and the shedding is shortest in the double mutant cells? Will that

14. Fig. 4C and D: Missing Standard deviations.

15. Fig. 4D: Please explain why the duration of shedding is longer in the myo51 ain1 mutant than the single mutants? This seems to contradict the hypothesis that the actin filaments are more stable in the ring of the double mutant cells.

16. Please indicate the number of independent biological repeats for each experiment.

Author Response

This manuscript by Tyree et al. studied the role of type V myosin Myo51 in cytokinesis using fission yeast as the model organism. By characterizing actinin Ain1 and Myo51 deletion mutants, the study demonstrated that contradictory to the authors’ hypothesis deletion of Myo51 results in more stable actin filament structures in the ain1 deletion mutant cells.

The paper describes an interesting interplay between a motor protein and a actin-crosslinker during cytokinesis. The discovery is very surprising. It uses a varies of unique approaches including “yeast motel” and super-resolution microscopy. The results shall be of interest to the field of cytokinesis. However, the reviewer has two major concerns.

We thank the reviewer for their thorough readthrough of our manuscript and for their thoughtful comments. We address each concern below. Based on the combined comments from multiple reviewers, we opted to change the order of the text. The section entitled Myo51 localizes to the inner layer of the contractile ring during constriction” can now be found at the end of the manuscript. Consequently, the order of the figures has been updated.

Major:

Q1: The genetic interaction between myo51 and ain1 mutants can be better characterized. For example, does the double mutant grow fast than the single ain1 mutant? Does the number of bi-nuclei cells among the myo51 ain1 mutant decrease, compared to the either myo51 or ain1 mutant? What about the septation indices of the single and double mutants? Lastly, how are the contractile assembly, maturation and constriction of the double myo51 ain1 mutant compared to the single ain1 mutant? Fig. 1C appear to address this last question partially, but this panel is hard to interpret in its current format and without the statistics. All the answers will clarify the genetic interaction between myo51 and ain1.

Answer:

Old Figure 1 is now Figure 4.

We never observed multiseptated or binucleated cells in ∆ain1, ∆myo51 or ∆ain1 ∆myo51 populations. We included this information within the text (see below text).

To help with the interpretation of the outcomes plots, swarm plots for timing of mitotic events were added to Supplemental Figure 1A-D and we properly referenced these new panels in the text. We also added the constriction rates and maturation duration for these genotypes in Supplemental Figure 1G-I and properly referenced these new panels in the text. We added the following text to the result section:

“Although ∆ain1 ∆myo51 cells exhibit a node clumping defect during contractile ring assembly, no multiseptated or binucleate cells were observed in the population (n = 150 cells) as observed in ∆ain1 and ∆myo51 cells suggesting that the contractile rings in those cells were still functional. In addition, the duration of maturation and the constriction rate of ∆ain1 ∆myo51 cells is similar to that of wild-type cell albeit significantly longer (Supplemental Figure 1G-I).”

Q2:  The phenotype of “shedding” can be characterized more quantitatively. It remains unclear how many fragments are being pushed out of the ring. The duration and size of the shredding shall also be quantified. Lastly, is there a correlation between shedding and the outcome of the contractile ring constriction?

Answer:

Old Figure 4 is now Figure 3.

We cannot unambiguously count the number of shedding fragments for each ring when cells are imaged flat. The duration of shedding can be found in Figure 3D. We measured the width of shedding in kymographs for wild-type, Δain1, Δmyo51 and Δain1 Δmyo51 cells. We have included this data in Supplemental Figure 1J and referenced it in the text.

“In ~70% of the cells, the fragments appeared more continuous with the contractile ring as they extended towards and often reached the lateral edges of the cell, highlighting the entire septa of both daughter cells (Figure 3A, B and F). These fragments were significantly longer in Δain1 Δmyo51 cells compared to wild-type, ∆ain1 and ∆myo51 cells (Supplemental Figure 1J). In addition, the shedding fragments were often so bright that we could not unambiguously distinguish the constricting contractile ring from the fragments.”

No multiseptated cells were seen in any of the genotypes measured. This indicates that exaggerated shedding does not impact the outcome of ring constriction. We added the following text to the result section:

“Although ∆ain1 ∆myo51 cells showed exaggerated shedding, no multiseptated ∆ain1 ∆myo51 cells were observed in the population suggesting that their contractile rings were generally functional and that premature and exaggerated shedding did not interfere with the constriction of the contractile ring.”

Minor:

Q3:  Author affiliations are not clear. Missing superscripts.

Answer: We have updated the manuscript with the appropriate superscripts.

Q4:    Typo: Line 34. Shall be cam1p.

Answer: The typo was corrected and now reads “Cam1p”.

Q5:      Does myo51 deletion alter the localization of the crosslinking protein Ain1 or Fim1? This may provide another potential explanation for the partial rescue of ain1 deletion mutant by the myo51 mutant.

Answer: It is not known if deleting myo51 impacts the localization of crosslinking proteins. It is known that deleting ain1 results in Fim1p localization to the contractile ring (Christensen et al., 2019 eLife), but it is uncertain whether this is further impacted by deleting myo51 as well. We will keep this experiment in mind in our future work.

We added a section of the discussion to include the potential impact of Fim1p to the ring of ∆ain1 ∆myo51 cells.

“Loss of Ain1p from the contractile ring results in the ectopic localization of fimbrin/Fim1p to the contractile ring as the two passive crosslinkers compete for actin filament binding [18,19]. The presence, the concentration and the timing of recruitment of Fim1p to the contractile ring of ∆ain1 ∆myo51 cells may help reconcile the complex interplay between Myo51 and Ain1p during cytokinesis. Future work will be necessary to determine whether and how Fim1p may be involved the clumping and shedding seen in ∆ain1 ∆myo51 cells.”

Q6: Fig. 1B What is the evidence that the actin cytoskeletal structures is disrupted in the presence of LatA for 6 mins?

Answer:

Old Figure 1 is now Figure 4.

We modified the text to clearly state that at such low concentration of LatA, the actin filament network partially breaks down. We clearly explain our rationale for using such low concentration as well. Nevertheless, this partial depolymerization is sufficient to release the Cdc15p-mCherry nodes and cause the dispersal of Myo51p-3GFP.

Q7 Fig. 1A: statistics of the distance between inner and outer contractile ring.

Answer:

Old Figure 1 is now Figure 4

We have included in the figure legend for Figure 4A that these distances correspond to the individual signals (see below for your convenience) in the accompanying micrographs (new text bolded), and thus statistical testing is not appropriate.

“Line graphs (right) depicting fluorescence intensity for both signals along the line, for each individual micrograph.”

Q8      Fig. 2A: please indicated percentage of the phenotypes (sever, mild) among the mutants.

Answer:

Old Figure 2 is now Figure 1.

We have indicated in the Figure 1F figure legends the percentage of severe and mild clumping among the whole population for each genotype as shown below (new text bolded):

F. Stacked bar chart of the percentage of cells expressing mEGFP-Myo2p that display mild (23% of Δain1 cells, 38% of Δain1 Δmyo51 cells amongst whole population) or severe (29% of Δain1 cells, 5% of Δain1 Δmyo51 cells amongst whole population) clumping phenotype. Asterisk, p < 0.05 by Pearson’s Chi-Square test.”

Q9      Fig. 2B: the ring in the top panel constricted earlier than the vertical line indicates.

Answer: We have corrected this alignment mistake.

Q10      Fig. 2C is very confusing. Please consider to split into multiple panels.

Answer:

Old Figure 2 is now Figure 1.

Old Figure 4 is now Figure 3.

We created swarm plots to represent these data differently and help with the confusion. These can be found in Supplemental Figures 1A-D. We did the same for the data shown in Figure 3E. The swarm plots representing the data from Figure 3E can be found in Supplemental Figure 1E and F.

Q11 Fig. 2E-G: missing standard deviations.

Answer:

Old Figure 2 is now Figure 1.

We created a new Table S2 and listed the means, standard deviations and sample size for all the graphs with swarm plots.

The data shown in Figure 1F is binary and standard deviations cannot be calculated for binary data.

Q12 Fig. 3: please add the LUT for fire pseudo-colored micrographs.

Answer:

Old Figure 3 is now Figure 2.

We have added the Fire LUT for all figure panels with fire pseudo-colored micrographs. We also included the Jet and Hot colormaps for the SMLM images in Figure 2F.

Q13 Fig. 4A: Missing LUT of the pseudo color. Please clarify when the “shedding” starts.

Answer:

Old Figure 4 is now Figure 3

We added the missing LUTs. We have increased the size of the arrows indicating the start of shedding for clarification.

Q14 Fig. 4C is confusing. Does the Y axis mean that the gap between the start of the ring constriction and the shedding is shortest in the double mutant cells? Will that

Answer:

Old Figure 4 is now Figure 3

Yes, the double mutant cells started shedding when the ring is less constricted, or closer to the start of constriction. We have clarified this in the following text (new text bolded).

“In ∆ain1 ∆myo51 cells, shedding started prematurely when the ring was 51% constricted (n = 51 cells) or an average circumference of 5.4 µm (Figure 3C).”

Q15. Fig. 4C and D: Missing Standard deviations

Answer:

Old Figure 4 is now Figure 3

We created a new Table S2 and listed the means, standard deviations and sample size for all the graphs with swarm plots.

Q16. Fig. 4D: Please explain why the duration of shedding is longer in the myo51 ain1 mutant than the single mutants? This seems to contradict the hypothesis that the actin filaments are more stable in the ring of the double mutant cells.

Answer:

Old Figure 4 is now Figure 3

We have clarified the possible explanation for the duration of shedding within the discussion section.

“Unlike during ring assembly, Myo51 may function as a crosslinker of actin filaments during constriction. The combined loss of two actin filament crosslinkers may destabilize the actin network and thus lead to premature and exaggerated shedding with an elongated shedding period.”

Q17. Please indicate the number of independent biological repeats for each experiment.

Answer:

Old Figure 2 is now Figure 1. Old Figure 4 is now Figure 3.

Biological variability in our genetically uniform population is captured in our sample size. New Table S3 lists all the statistical test results for Figures 1C and 3E. Where missing (Figures 1C and 3E), we added the range of n-values in the appropriate figure legends as follows (new text bolded):

Figure 1C: “C. Outcome plot of the timing of assembly completion (n=38-46), onset of constriction (n=34-42), clumping start (n=37-43) and clumping end (n=37-43) in cells expressing mEGFP-Myo2p.

Figure 3E: “E. Outcome plot showing the timing of shedding start (n=39-54) and disassembly completion (n=30) in cells expressing mEGFP-Myo2p.

Round 2

Reviewer 2 Report

In the revised manuscript, the beginning of the result section has become easy to follow. However, the manuscript is still so unfocused that it requires fundamental reconstitution before publication. The reduction of the node clumps in Δmyo51Δain1 is interesting, but the authors proceeded to address the other issues without any further efforts to elucidate its mechanism, which make the manuscript descriptive. 

Disassembly of the contractile ring by shedding is also interesting and seems novel. The authors observed different behaviors between the outer and inner components of the contractile ring in the process, and found that Myo51 and Ain1p are involved in shedding. I think these results are more valuable than the cooperation of Myo51 and Ain1p.

Regarding the method section, I indicated the concerns about the objectivity in some data in the first reviewing, and found that some analyses actually lacked objectivity since the authors just copied the corresponding parts in the results as their methods.

Section 2.5

‘intense signal separated by dim signal’, ‘mild contrast’, and ‘bright enough to obscure’ should be objectively determined by intensity values.

Author Response

Reviewer 2

In the revised manuscript, the beginning of the result section has become easy to follow. However, the manuscript is still so unfocused that it requires fundamental reconstitution before publication. The reduction of the node clumps in Δmyo51Δain1 is interesting, but the authors proceeded to address the other issues without any further efforts to elucidate its mechanism, which make the manuscript descriptive. 

Disassembly of the contractile ring by shedding is also interesting and seems novel. The authors observed different behaviors between the outer and inner components of the contractile ring in the process, and found that Myo51 and Ain1p are involved in shedding. I think these results are more valuable than the cooperation of Myo51 and Ain1p.

Regarding the method section, I indicated the concerns about the objectivity in some data in the first reviewing, and found that some analyses actually lacked objectivity since the authors just copied the corresponding parts in the results as their methods.

ANSWER: We thank the reviewer for their honest opinions on this work. Although all results from our experiments do not perfectly fit into a seamless narrative, we reported the unaltered results to the experimental approaches used to attempt to understand the function of Myo51 during cytokinesis. 

Section 2.5

‘intense signal separated by dim signal’, ‘mild contrast’, and ‘bright enough to obscure’ should be objectively determined by intensity values.

ANSWER: The severity of the clumping phenotype was first determined qualitatively using the descriptors mentioned in the method section (pasted below for your convenience). To confirm the interpretation that severe clumps contained more nodes than the clumps in cells with a mild clumping phenotype, we measured the fluorescence intensity of the clumps (Figure 1G).  In general, the trend showed that rings that we qualitatively categorized as “severe” had clumps with a higher mean fluorescence intensity.

“A ring with mild clumping refers to a contractile ring with no gaps in signal but with localized bright and dim spots around the ring. A ring with severe clumping was defined by a discontinuous fluorescent signal with large gaps in the ring or stretches of intense mEGFP-Myo2p signal separated by dim signal. A ring with a moderate clumping phenotype exhibited mild contrast between stretches of brighter and dimmer fluorescent signal.”

Reviewer 3 Report

The authors have answered most questions raised by the reviewer in the last round of review. 

The only question that remains unanswered is the growth of the ain1 myo51 double mutant. Please compare the growth of ain1 deletion, myo51 deletion and the ain1 myo51 double mutant at various temperatures. This can be done through either ten-fold dilution series or growth curve. This result can help determine whether the genetic interaction between the ain1 and myo51 mutants have any effects on cell proliferation.  

Author Response

Reviewer 3

The only question that remains unanswered is the growth of the ain1 myo51 double mutant. Please compare the growth of ain1 deletion, myo51 deletion and the ain1 myo51 double mutant at various temperatures. This can be done through either ten-fold dilution series or growth curve. This result can help determine whether the genetic interaction between the ain1 and myo51 mutants have any effects on cell proliferation.  

Answer: We performed a dilution assay for wild-type control, ∆ain1, ∆myo51 and the double deletion ∆ain1 ∆myo51 strains. The dilution assays were grown at 25, 32 and 36˚C. All strains grew equally well at all temperatures tested with no noticeable difference between the genotypes. The new figure panel can be found in Supplemental Figure 1K. The results section, methods and figure legends were update to include this new information.